Evolutionary and ecological processes influencing chemical defense variation in an aposematic and mimetic Heliconius butterfly

Mattila Anniina L. K. anniina.mattila@helsinki.fi 1 2 7
Jiggins Chris D. 3
Opedal Øystein H. 4
Montejo-Kovacevich Gabriela 3
Pinheiro de castro Érika C. 3
McMillan W. Owen 5
Bacquet Caroline 6
Saastamoinen Marjo 1 2
1 Research Centre for Ecological Change, Organismal and Evolutionary Biology Research Programme, University of Helsinki , Helsinki , Finland
2 Helsinki Life Science Institute, University of Helsinki , Helsinki , Finland
3 Department of Zoology, University of Cambridge , Cambridge , United Kingdom
4 Department of Biology, Lund University , Lund , Sweden
5 Smithsonian Tropical Research Institute , Gamboa , Panama
6 Universidad Regional Amazónica de Ikiam , Tena , Ecuador
7 *Current affiliation: Finnish Museum of Natural History (LUOMUS), University of Helsinki , Helsinki , Finland
Garant Dany
Electronic publication date: 2021 Jun 18
Publication date: 2021
Volume: 9
Electronic Location ID: e11523
Received 2021 Feb 24; Accepted 2021 May 5
Copyright: ©2021 Mattila et al.
Copyright year: 2021
Copyright holder: Mattila et al.
License: This is an open access article distributed under the terms of the Creative Commons Attribution License, which permits unrestricted use, distribution, reproduction and adaptation in any medium and for any purpose provided that it is properly attributed. For attribution, the original author(s), title, publication source (PeerJ) and either DOI or URL of the article must be cited.
License URL: https://creativecommons.org/licenses/by/4.0/

Keywords: Chemical defenses, Aposematism, Mimicry, Evolvability, Maternal effects, Environmental gradients, Heliconius, Passiflora, Cyanogenic glucosides

Funding: Academy of Finland 286814 ERC 339873 NERC Doctoral Training Partnership NE/L002507/1 Spanish Agency for International Development Cooperation 2018SPE0000400194 Financial support was provided by the Academy of Finland (Grant no. 286814 to Anniina LK Mattila), ERC (SpeciationGenetics 339873 to Chris D Jiggins), NERC Doctoral Training Partnership (NE/L002507/1 to Gabriela Montejo-Kovacevich), and Spanish Agency for International Development Cooperation (AECID, grant number 2018SPE0000400194 to Caroline Bacquet). Open access was funded by the Helsinki University Library. The funders had no role in study design, data collection and analysis, decision to publish, or preparation of the manuscript.

==============================
Chemical defences against predators underlie the evolution of aposematic coloration and mimicry, which are classic examples of adaptive evolution. Surprisingly little is known about the roles of ecological and evolutionary processes maintaining defence variation, and how they may feedback to shape the evolutionary dynamics of species. Cyanogenic Heliconius butterflies exhibit diverse warning color patterns and mimicry, thus providing a useful framework for investigating these questions. We studied intraspecific variation in de novo biosynthesized cyanogenic toxicity and its potential ecological and evolutionary sources in wild populations of Heliconius erato along environmental gradients, in common-garden broods and with feeding treatments. Our results demonstrate substantial intraspecific variation, including detectable variation among broods reared in a common garden. The latter estimate suggests considerable evolutionary potential in this trait, although predicting the response to selection is likely complicated due to the observed skewed distribution of toxicity values and the signatures of maternal contributions to the inheritance of toxicity. Larval diet contributed little to toxicity variation. Furthermore, toxicity profiles were similar along steep rainfall and altitudinal gradients, providing little evidence for these factors explaining variation in biosynthesized toxicity in natural populations. In contrast, there were striking differences in the chemical profiles of H. erato from geographically distant populations, implying potential local adaptation in the acquisition mechanisms and levels of defensive compounds. The results highlight the extensive variation and potential for adaptive evolution in defense traits for aposematic and mimetic species, which may contribute to the high diversity often found in these systems.

Introduction

Chemical defenses are a common means for animals to gain protection against predators. Defensive chemicals can be acquired from the larval or adult diet (sequestered) and/or biosynthesized de novo by the organism (Bowers, 1992; Opitz & Müller, 2009; de castro et al., 2019; de castro et al., 2021). Chemical defenses are often coupled with bright color patterns, which local predators learn to associate with toxicity or unpalatability of the prey, a phenomenon known as aposematism (Ruxton et al., 2018). This can also lead to mimicry among co-occurring prey species, one of the best studied examples of Darwin’s theory of evolution by natural selection and the subject of one of evolution’s oldest mathematical models (Müller, 1879). Because palatability can vary along a continuum (Turner, 1984; Speed, 1999; Speed et al., 2012; Arias et al., 2016; Prudic et al., 2019), mimicry can take several forms; Batesian, where a palatable mimic exploits an unpalatable model, Müllerian, in which unpalatable species copy one another for mutual benefit (the relationship can be mutualistic even with unequal defenses; Rowland et al., 2007), or Quasi-Batesian, where a less well protected species acts in a parasitic manner, diluting the protection of the better defended species (Turner, 1987; Speed, 1993; Speed, 1999; Mallet, 1999).

These different mimicry relationships all imply predictable differences in toxicity among mimetic species and individuals (Turner, 1987; Speed, 1999), and yield different predictions about the origins of diversity in warning coloration and mimicry, which is often greater than predicted by classic aposematic theory (Joron & Mallet, 1998; Speed, 1999; Briolat et al., 2019). For example, alternation between different mimicry types could influence patterns of convergent vs. advergent selection (where selective pressures cause phenotypic convergence of one species on another, but not vice versa) (Mallet & Joron, 1999; Mallet, 1999; Speed, 1999). However, there are many open questions related to the roles of ecological and evolutionary processes shaping chemical defense variation in aposematic and mimetic species (Mallet & Joron, 1999; Speed et al., 2012; Ruxton et al., 2018; Briolat et al., 2019). One outstanding current question is “what drives variation in defensive toxins within and among prey populations?”

Extensive variation in defensive toxins within species and across mimicry rings has been demonstrated repeatedly (Speed et al., 2012). Indeed, unpalatability levels of mimic species can range from equal to very uneven (Ritland & Brower, 1991; Bowers, 1992; Arias et al., 2016). Potential sources of this variation include host-plant chemistry, which often varies with the environment and is influenced by resource availability (Massad et al., 2011). Host-plant chemistry can be a major source of intraspecific variation particularly in sequestered defensive compounds (Bowers, 1992; Opitz & Müller, 2009), as seen in many butterfly species (Gardner & Stermitz, 1988; Camara, 1997; Hay-Roe & Nation, 2007). Detoxification of host plant defensive chemicals can also negatively affect an individual’s resource budget, leading to variation in investment in defenses (Lindstedt et al., 2010; Reudler et al., 2015). Similarly, species that have evolved the ability to biosynthesize defensive compounds often vary extensively in their defensive chemicals. In this case, the origin of both inter- and intraspecific variation is less clear (Ruxton et al., 2018; Burdfield-Steel et al., 2019; Zvereva & Kozlov, 2016). Sources of variation may include biotic factors, such as the energy and resources available for toxin synthesis, interactions with life-history traits including age and reproductive stage, and general condition as assayed by e.g., immunological status (Bowers, 1992; Smilanich et al., 2009; Zvereva & Kozlov, 2016). This variation could simply be maintained by drift if toxicity profiles are selectively neutral, for example if acquiring and maintaining defenses does not incur costs for the individual, and variation in defense level does not affect predator behavior (Speed et al., 2012; Briolat et al., 2019).

In those species that sequester and those that synthesize defensive compounds, there are also several potential adaptive explanations for the persistence of variation in nature (Speed et al., 2012). The selection pressures acting on chemical defense traits are expected to be complex, as chemically defended herbivores are often involved in close coevolutionary arms-races with their host plants, predators and parasitoids (Turner, 1984; Ruxton et al., 2018). Variation in defenses can be maintained by frequency-dependent predator-mediated selection (Skelhorn & Rowe, 2005), or the costs of producing and maintaining toxicity resulting in frequency- or density-dependent selection in resource optimization and warning-signal honesty (Blount et al., 2009; Blount et al., 2012; Speed et al., 2012; Arenas, Walter & Stevens, 2015). Defense-related costs could lead to energetic trade-offs (Bowers, 1992; Fordyce & Nice, 2008; Lindstedt et al., 2020) and automimicry (the occurrence of palatable “cheaters” in a chemically defended population; Brower, Pough & Meck, 1970; Speed et al., 2012). Evolutionary explanations for chemical defense variation are only plausible if such variation is genetically determined, and sufficient genetic variation is available for selection to act on. Therefore, knowledge about the heritability and evolvability of chemical defense traits is fundamental in evaluating the roles of evolutionary vs. ecological processes in the origins of the extensive variation observed in these traits. Overall, empirical data unraveling the patterns and origins of chemical defense variation are needed to help build a more complete understanding of the evolution and diversification of aposematic and mimetic systems.

Here, we investigate aspects of chemical defense variation in neotropical Heliconius butterflies. With their incredible diversity in wing coloration, they are a well-known example of aposematism and mimicry (Merrill et al., 2015; Jiggins, 2017), and provide a powerful study system in which to assess the role of ecology and evolution in chemical defense variation within species and populations. Heliconius larvae feed exclusively on passion vines (Passiflora), with which they have radiated and speciated through a coevolutionary arms race (Benson, Brown & Gilbert, 1975; de castro et al., 2018). Almost 30 different cyanogenic glucoside structures have been reported in the Passiflora genus (de castro et al., 2019) and only the most simple of these compounds can be sequestered by Heliconius larvae (Engler, Spencer & Gilbert, 2000; de castro et al., 2019). Heliconius larvae and adult butterflies can additionally biosynthesize the cyanogenic compounds linamarin and lotaustralin de novo from aliphatic amino acids (Nahrstedt & Davis, 1983). Although biosynthesis is not uncommon in chemically defended animals (Bowers, 1992; Ruxton et al., 2018), it is uncommon among chemically defended Lepidoptera (Nishida, 2002). There is substantial variation in the amounts and types of cyanogens among different Heliconius species (Arias et al., 2016; Cardoso & Gilbert, 2013; de castro et al., 2019; Engler-Chaouat & Gilbert, 2007; Sculfort et al., 2020). Some of this is explained by the level of diet specialization, which is a major determinant of the relative roles of cyanogen sequestration from plants and de novo biosynthesis (Engler-Chaouat & Gilbert, 2007; de castro et al., 2019; de castro et al., 2021). The level of sequestered defenses is also known to be affected by variation in the chemical content of the host plants (Engler-Chaouat & Gilbert, 2007; Hay-Roe & Nation, 2007), whereas the sources of variation in biosynthesis are less well understood.

We studied ecological and evolutionary aspects of intraspecific variation in biosynthesized cyanogenic toxicity in Heliconius erato using a combination of extensive field collections and common-garden rearing experiments. Heliconius erato provides an interesting study subject in terms of variation in chemical defenses, warning coloration patterns and mimicry relationships. It is found in ecologically diverse habitats across its large geographic range extending from northern Argentina to Mexico, and has a diverse array of color forms associated with local color pattern mimicry rings, the size of which vary from a few comimetic species in Central America to up to a dozen in the Amazon (Supple et al., 2013; Jiggins, 2017). The color forms are region-specific: similar between comimetic species within any one area, but changing in concert across the geographic range (Supple et al., 2013; Jiggins, 2017). Our field collections exploit two strong environmental gradients within this geographic range: a rainfall gradient with a five-fold difference in rainfall, and a steep altitudinal gradient in the Andes. We use these data to tackle three questions: (1) What is the extent of natural variation in toxicity within and among populations of H. erato? (2) Can some of this variation be explained by environmental gradients or highly divergent habitats and mimicry environments across a wider geographic scale? (3) How much of the variation in biosynthesized chemical defenses is heritable and evolvable, and how much is governed by environmental (particularly dietary) variance? By examining our results in the context of the well-studied mimicry system of Heliconius butterflies, we discuss the potential wider implications of our findings for aposematic and mimetic systems.

Methods

Study sites and field collection of butterflies and host plants

Heliconius erato demophoon and its typical host plant P. biflora were collected in Panama along a steep rainfall gradient (the Smithsonian Tropical Research Institute Physical Monitoring Program; https://biogeodb.stri.si.edu/physical_monitoring/) from three populations (Wet, Intermediate and Dry study sites; Fig. 1A, File S4), and H. erato lativitta from High and Low altitude sites on the Eastern slope of the Andes in Ecuador (Fig. 1A, File S4). Research and collecting permits were granted by the Ministerio de Ambiente, Republica de Panama (Permit SE/AP-21-16) and the Ministerio del Ambiente, Ecuador (under the Contrato Marco MAE-DNB-CM-2017-0058). Butterflies were caught using a hand net and transported live to the laboratory. All individuals caught from the Panamanian study areas (n = 92, of which 34, 28 and 30 individuals were caught from Dry, Intermediate and Wet sites, respectively) were separated by sex, weighed, and the body excluding wings and one half of the thorax was preserved in 1 ml 100% MeOH (methanolic preservation stabilizes cyanogenic glucosides (Gleadow et al., 2011), and was used with both butterfly and plant samples) on the day of capture (males) or after being allowed to oviposit in an insectary (females, see File S4). At the Ecuadorian sites, fertilized females were brought to a common garden environment and kept in separate cages. Eggs were collected daily, and larvae raised in individual containers all through development, placed in randomly assorted positions in common garden insectary conditions. The thorax of 21 F1 individuals from each altitude (total n = 42) were sampled in 1 ml 100% MeOH. We do not expect the sampled body part to influence estimates of cyanogenic glucoside compound concentrations, as these compounds are located in the hemolymph, and are thus expected to be evenly distributed throughout the body (de castro et al., 2020).

Figure 1 Natural variation and evolutionary potential of biosynthesized cyanogenic toxicity in Heliconius erato.

(A) Top panel: Locations of the natural study populations in Central and South America (Panama and Ecuador, respectively), with a more detailed map of study areas along the rainfall gradient in Panama. Bottom panel: Experimental setups for common garden populations of H. erato and host plant Passiflora biflora. (B) Variation among natural populations in total cyanogenic and total biosynthesized cyanogenic (CNglc) toxin concentrations in P. biflora and H. erato, respectively. The boxplots show the range, the first and third quartiles and the median. Small-case letters indicate statistically significant differences between populations (Tukey HSD P < 0.05). (C) Zero-truncated kernel density distributions of total biosynthesized cyanogenic toxicity in natural populations of H. erato in Panama. (D) Top panel: Variance component estimates of H. erato biosynthesized cyanogenic toxicity (with whiskers representing S.E.) based on pedigree data (see panel A and Table 2). Bottom panel: The histogram shows the distribution of repeatability estimates in 1,000 parametric bootstraps with brood identity as the grouping factor. The blue dot indicates the repeatability estimate, and whiskers represent the 95% CI.

Passiflora biflora leaf samples (one to four leaves pooled per plant individual, see File S4) were collected from the Panama Dry (n = 20), Intermediate (n = 11) and Wet (n = 20) study areas (Fig. 1A), and preserved fresh in 1ml 100% MeOH. In addition, 50 cm cuttings, all from different P. biflora individuals, were collected from the Dry (n = 12) and Wet (n = 12) study areas for greenhouse cultivation.

Host plant greenhouse cultivation and treatments

Passiflora biflora were cultivated in greenhouses (75% relative humidity; 8-20 h: 25 °C, light; 20-8 h: 20 °C, dark) in a soil mixture (50% compost, 20% coir, 15% perlite, 15% sand or gravel) (see File S4 for details on cultivation methods and watering treatments). Standard (std)-treated P. biflora were used as oviposition plants for the H. erato parental generation and larval diet for the F1 families not included in feeding treatments (see below). The quality of host plants in H. erato feeding treatments was manipulated with watering treatments, motivated by the importance of water availability for many cyanogenic plants (Gleadow & Woodrow, 2002; Hayden & Parker, 2002; Liang, 2003; Vandegeer et al., 2013). The P. biflora cuttings collected from the Dry and Wet study areas in Panama were cultivated in a full-factorial combination of origins and water treatments, resulting in four host plant types. Plants in the dry treatment were kept at 75% relative humidity and watered two to three times weekly with about 1/3 of the water volume compared to the wet treatment (constant water availability, relative humidity up to 100%), such that the surface soil was allowed to dry and the leaves slightly drooped between watering events. The dry treatment thus aimed to mimic conditions of drought stress.

Butterfly rearing, breeding design and feeding treatments

Laboratory H. erato populations descendent of the wild Dry population were established in greenhouse conditions (see Supplementary File 4 for details on rearing methods). Larvae were reared on std-type P. biflora. All butterflies were individually marked, and were allowed to fly, mate, and oviposit freely in 2 ×  2 × 2 m mesh cages with potted std-type P. biflora and a standardized diet of sugar solution supplemented with amino acids provided ad libitum. The butterfly cages were observed throughout the day for mating pairs, and mated females were moved to individual 2 ×  2 × 2 m mesh cages and allowed to oviposit on std-type P. biflora. Eggs of 14 females (eight mated with a known father, average no. of full-sibs/family = 8, see File S2) were collected, and hatched larvae were reared in full-sib groups fed on std-type P. biflora until pupation.

The effect of larval dietary cyanogen content and quality on adult butterfly biosynthesized toxicity was studied with feeding treatments. We expected larval dietary quality to affect butterfly toxicity due to potential variation in the acquisition of nitrogenous resources for cyanogen biosynthesis (Cardoso & Gilbert, 2013), possible detoxification costs (Lindstedt et al., 2010; Reudler et al., 2015) and the general importance of larval diet for fitness traits in holometabolous insects (e.g., Boggs & Freeman, 2005). The offspring of six mated pairs (both parental identities known, average no. of full-sibs/family = 29, File S2) were included in the feeding treatments, where eggs from each mother were divided equally into four groups and transferred onto each of the four P. biflora treatment types. Individuals of the F1 generation were collected within three days following emergence (unmated and unfed), weighed, and preserved in 100% MeOH (body excluding wings and one-half of thorax).

Butterfly toxicity analyses with 1H-NMR

The concentrations of the two cyanogenic compounds biosynthesized by Heliconius larvae and adults, linamarin and lotaustralin, were analyzed from the butterfly samples using nuclear magnetic resonance (1H-NMR) with Bruker Avance III HD NMR spectrometer (Bruker BioSpin, Germany) (see File S4 for details on sample extraction and 1H-NMR methods). The data were processed and analyzed using Bruker TopSpin software (versions 3.2.6; 2014 and 3.5.6; 2016). The peaks of cyanogen compounds were identified based on comparison of sample 1H spectra with the 1H spectra of authentic reference samples of linamarin and lotaustralin. Note that we analyzed only the biosynthesized cyanogenic compounds (not sequestered cyanogenic compounds which differ in chemical structure from the two biosynthesized compounds linamarin and lotaustralin, nor other chemicals potentially influencing unpalatability in Heliconius; see (Jiggins, 2017)), and Heliconius “cyanogenic toxicity” or “toxicity” refer here to biosynthesized cyanogenic toxicity. Our study species has been considered generalist or oligophagous on Passiflora, largely acquiring its cyanogenic toxicity through de novo biosynthesis of cyanogens (Engler-Chaouat & Gilbert, 2007; Merrill et al., 2013), and the cyanogen profile of its main host plant P. biflora, also used in our experimental setup, consists of complex compounds which Heliconius larvae are unable to sequester (Engler-Chaouat & Gilbert, 2007; de castro et al., 2019).

All statistical analyses of the study were undertaken using R 3.4.4 (R Core Team, 2018). The values of cyanogen concentrations in the wild-collected samples were normalized with a square-root-transformation before applying ANOVA to test for population differences. Distributions of cyanogen traits were also explored by inspection of truncated weighted density distributions (accounting for the skewness of data towards near-zero values) using the “sm” R package (Bowman & Azzalini, 2014). For the analyses using common-garden data, the data were normally distributed and untransformed values were used.

Host plant toxicity and quality analyses

The cyanogenic content of the Passiflora biflora samples was analyzed using liquid chromatography-mass spectrometry (LC-MS/MS) with Agilent 1100 Series LC (Agilent Technologies, Germany) hyphenated to a Bruker HCT-Ultra ion trap mass spectrometer (Bruker Daltonics, Germany) following the procedure of de castro et al. (2019) (see File S4 for details on sample extraction, spectrometry and analyses). Mass spectral data were analyzed with the native data analysis software (Compass DataAnalyses, Bruker Daltonics). The cyanogen compounds of P. biflora were detected and quantified following de castro et al. (2019).

Plant quality traits assumed to be proxies of nutritional value, including nitrogen/carbon ratio (a proxy of protein/carbohydrate-ratio), were measured from the greenhouse-cultivated P. biflora treatment types (average number of individual leaves sampled per P. biflora treatment type: n = 63) with Dualex® Scientific+ leaf clip meter (Cerovic et al., 2012; File S4). We tested for differences in cyanogen levels and plant quality traits between the treatment groups in an ANOVA including plant origin, treatment and their interaction.

Estimation of variance components, heritability, evolvability and maternal effects in Heliconius biosynthesized cyanogenic toxicity

The concentrations of the two tested biosynthesized cyanogens, linamarin and lotaustralin were strongly correlated with each other (r =0.897). Cyanogen concentration was not correlated with adult body mass in either sex (F1,193 = 0.52, P = 0.470 for total cyanogen concentration; mass data available for n = 196 individuals). Adult mass also did not differ between the feeding treatments (F4,186 = 0.88, P = 0.480; model including feeding treatment, sex and their interaction), and was not included as a variable in further analyses.

Variance components of Heliconius erato biosynthesized cyanogenic toxicity were analyzed based on the two-generation full-sib/half-sib breeding design (Lynch & Walsh, 1998a) by fitting linear mixed models applying the “animal model” framework (Kruuk, 2004; Wilson et al., 2010), in which the full pedigree (relatedness matrix) is incorporated as an individual-level random effect. The models were fitted using ASReml-R Version 4 (Butler et al., 2017), an R package that fits linear mixed models using Residual Maximum Likelihood (REML) estimation. The two-generation pedigree consisted of 20 broods and their parents, and included mother-offspring, father-offspring, full-sibling and paternal half-sibling relationships (n = 322 individuals, File S2). Of the 20 broods, six were included in the feeding treatments receiving four different P. biflora types as larval diet (File S2). Individuals of the parental generation were assumed to be unrelated. With the materials available, we maximized our power to detect genetic variation by generating as many family groups as possible. Because the number of families was limited, we also analyzed a large number of offspring per family, which allowed precise estimates of family means. Because our breeding design resulted mostly in full-sib families, we have limited power to distinguish additive-genetic from other genetic variance components (dominance, epistasis, maternal/paternal effects). We thus consider our genetic-variance estimate as a ‘broad-sense’ genetic variance, although the estimate approaches the narrow-sense estimate more than a simple full-sib design due to the parent–offspring and half-sib relations in the breeding design (Lynch & Walsh, 1998a).

Animal models were fitted separately for the three biosynthesized cyanogenic toxicity traits (concentration of linamarin, lotaustralin and total cyanogens). The primary models (referred to as “Animal Model1”) included the individual breeding value as a random effect, and the fixed effects of sex and feeding treatment (variance explained by fixed effects computed using the method of de Villemereuil et al. (2018); see also File S4). “Animal Model2” included additionally the random effect of maternal identity. We computed broad-sense heritabilities (H2) as the proportion of the total phenotypic variance (VP) explained by the (broad-sense) genetic variance component (VG). When calculating heritabilities we included the variance due to sex in the total phenotypic variance, but we did not include the variance due to the treatment because this component of the variance is unlikely to represent natural variation (de Villemereuil et al., 2018). Heritabilities are useful measures of the resemblance between offspring and parents, but are poor standardized measures of evolutionary potential (Hansen, Pélabon & Houle, 2011). As a measure of evolutionary potential, we therefore computed mean-scaled evolvabilities (eμ) as eμ = VG/µ2, where µ is the trait mean (of common garden brood data trait values), which gives the expected percent change in the trait mean per generation under a unit strength of selection (Hansen & Houle, 2008; Hansen, Pélabon & Houle, 2011).

Given that our design included mostly full-sib groups, we obtained an additional estimate of the component of toxicity variance explained by brood by estimating the intra-class correlation coefficient (ICC), i.e., repeatability, with a linear mixed model approach. For this, we implemented the “rptR” package (Stoffel, Nakagawa & Schielzeth, 2017), with brood identity as the grouping factor, a Gaussian distribution and 1000 parametric bootstraps to quantify uncertainty.

To further explore the role of maternal contributions to the inheritance of cyanogenic toxicity, we estimated maternal-effect coefficients following Falconer’s dilution model, which assumes that offspring phenotype is a direct, linear function of maternal phenotype, beyond the contribution of maternally transmitted genes (Lande & Price, 1989; Lynch & Walsh, 1998b; Walsh & Lynch, 2018b). Thus, the interpretation of this ‘maternal effect’ is related to specific measured traits (here, cyanogenic toxicity), and is different from the typical formulation of ‘maternal environmental effects’ (influence of the maternal environment on offspring traits). We estimated maternal effect coefficients (m) as the difference in slope between a mother-offspring (mo) and a father-offspring (fo) regression, i.e., m = β mo − β fo (Lande & Price, 1989; Lynch & Walsh, 1998b). We fitted the parent–offspring regressions as linear mixed models with the lme4 R package (Bates et al., 2014), including family as a random factor and maternal or paternal toxicity phenotype as explanatory variables of offspring toxicity phenotype.

Results

First, we investigated the ecological origins of variation in defensive traits by studying toxicity of butterflies along steep gradients in rainfall and altitude (Fig. 1A). Toxicity levels were similar between populations within the gradients, and variation in butterfly toxicity was not detectably associated with toxicity variation of the major host plant species along the rainfall gradient (Fig. 1B). A comparison of geographically distant natural populations of H. erato from Panama and Ecuador showed that populations of the same species can have substantially different levels of biosynthesized toxicity (Fig. 1B). Furthermore, within natural H. erato populations sampled along the rainfall gradient in Panama, the distribution of biosynthesized cyanogenic toxicity was consistently skewed, so that a large proportion of individuals had a noticeably low level of toxicity compared with a smaller proportion of highly toxic individuals within all sampled populations (Fig. 1C). Finally, we partitioned the variation in toxicity into genetic and environmental components, with particular focus on the variance originating from diet. Common-garden pedigree data comprising 20 broods and more than 300 Heliconius erato individuals revealed a moderate but detectable component of variance explained by brood identity (around 12% of the total variance; Fig. 1D). However, further analyses suggested that the among-brood variance may not primarily represent additive genetic variance but rather be driven by maternal environmental or genetic influences on offspring toxicity phenotype. In contrast, nutritional quality indicators and cyanogenic toxicity of the host plant Passiflora biflora, which varied in response to water availability treatments, had no effect on the biosynthesized toxicity level of the adult butterflies that had fed on these plants as larvae. Together, these results demonstrate high levels of intraspecific variation that is not detectably driven by the ecological factors studied here. They also indicate complex modes of inheritance in this chemical defense trait of an aposematic and mimetic Heliconius butterfly.

Wide natural variation in toxicity within populations and between regions, but not along environmental gradients

The distribution of biosynthesized cyanogen levels, as estimated from the concentrations of biosynthesized compounds linamarin and lotaustralin, was similar across natural populations of H. erato sampled along an environmental rainfall gradient in Panama (Fig. 1C). A large proportion of individuals had low toxicity values (e.g., 40% of individuals had total cyanogen concentrations falling within the lower 25% of toxicity values), while a small proportion of individuals had a considerably higher toxicity level (e.g., 5% of individuals had total cyanogen concentrations within the highest 25% of toxicity values). Total cyanogen concentration and concentration of linamarin did not differ among the three Panamanian populations (Fig. 1B, Table 1, File S1). The concentration of lotaustralin differed among populations (F2,89 = 7.97, P < 0.001), with the Intermediate population having higher concentrations than the two populations at the opposite extremes along the rainfall gradient (File S1).

The total cyanogen levels of the host plant Passiflora biflora differed among the natural populations along the rainfall gradient in Panama (F2,48 = 3.65, P = 0.034; Fig. 1B, Table 1). The Dry population had substantially lower cyanogen levels compared with both the Intermediate and Wet populations (Fig. 1B, Table 1). In contrast, levels of cyanogen did not vary substantially across populations of H. erato (Fig. 1B, Table 1, File S1).

The populations along the altitudinal gradient in Ecuador did not differ in biosynthesized cyanogenic toxicity from each other, but differed markedly from their Panamanian conspecifics (Fig. 1B, Table 1). Most Ecuadorian samples lacked both linamarin and lotaustralin (File S1). Linamarin was detected in only 29% of samples, most of which originated from the high-altitude site (75% of those with any signal), and the average biosynthesized cyanogen level in these samples was very low (0.063%; Fig. 1B, Table 1, File S1), which is below the minimum level observed in the Panamanian populations.

Biosynthesized toxicity does not vary with diet but varies among broods

The level of biosynthesized cyanogenic toxicity varied two-fold among the 20 common-garden reared broods (range of brood averages =0.48%–0.88% of dry mass, Fig. 2A). Phenotypic variance components of cyanogenic toxicity estimated within an “animal model” framework (Kruuk, 2004; Wilson et al., 2010), which included the full pedigree as an individual-level random effect and fixed effects of sex and feeding treatment (Model1), revealed a statistically significant genetic variance component (VG) for total biosynthesized cyanogen concentration level (Fig. 1D, Table 2, File S1), corresponding to a broad-sense heritability (H2) of 0.115 (H2 (linamarin) =0.139, H2 (lotaustralin) =0.067; Table 2, File S1). We confirmed the presence of detectable among-brood variance in a simpler brood-level analysis, in which brood identity explained 12.5% of the variation in biosynthesized cyanogen concentration across common-garden reared broods (repeatability R = 0.125, SE =0.059, CI = [0.015, 0.245], P = 6.02e−05). Among-brood variance could originate from several sources including (1) additive genetic variance, (2) maternal environmental or genetic effects, and (3) shared environment, and below we explore these options further.

Table 1 Total cyanogen concentrations (mean, standard deviation (SD) and P-values of Tukey HSD pairwise comparisons) in natural populations of Heliconius erato and Passiflora biflora sampled along environmental gradients in Panama and Ecuador.

Statistically significant pairwise comparisons are marked in bold. See File S1 for full results table with different cyanogen compounds shown separately.

		H. erato total cyanogens %	P. biflora total cyanogens (µg/mg)	
Country	Population	Mean	SD	Dry	Int	Wet	Low	High	Mean	SD	Dry	Int	Wet	
Panama	Dry	1.064	0.709						11.850	10.073				
Panama	Intermediate	1.271	0.631	0.432					21.544	13.586	0.038			
Panama	Wet	0.907	0.521	0.905	0.098				18.005	7.953	0.146	0.627		
Ecuador	Lowland	0.011	0.028	<0.001	<0.001	<0.001			 					
Ecuador	Highland	0.026	0.033	<0.001	<0.001	<0.001	0.936		 					

Table 2 Summary of variance components of Heliconius biosynthesized cyanogenic (CNglc) toxicity estimated with REML animal models.

V mat, V trm, V sex and V R are the variance components for maternal effects, host plant feeding treatment, sex, and residual variance, respectively. See File S1 for full model results.

					Genetic variance component, heritability and evolvability	Other variance components (V/VP)	Total phenotypic variance	
Model type	Cyanogen
trait	Trait mean (µ)	No. Obs.	No. broods	VG	P VG	H2 (VG/VP)	eμ (VG/µ2); %	Vmat/ VP	Vtrm/ VP	Vsex/ VP	VR/ VP	VP	
1	Total CNglc %	0.772	322	20	0.0093	0.037	0.115	1.555	 	0.011	0.011	0.874	0.080	
1	Linamarin %	0.577	322	20	0.0063	0.020	0.139	1.887	 	0.001	0.013	0.848	0.045	
1	Lotaustralin %	0.195	322	20	0.0004	0.130	0.067	1.039	 	0.008	0.004	0.929	0.006	
2	Total CNglc %	0.772	322	20	1.0e−7	1.000	1.4e−6	1.7e−5	0.102	0.043	0.002	0.895	0.073	

Figure 2 Variation of biosynthesized cyanogenic toxicity in common-garden broods of Heliconius erato.

(A) Variation in total biosynthesized cyanogen (CNglc) concentration (% dry mass) among full-sibling broods of Heliconius erato by increasing maternal CNglc % (one pair of full-sib broods are paternal half-siblings, in grey). The boxplots show the range, the first and third quartiles and the median. (B) Total biosynthesized CNglcs concentration (% dry mass) in newly-emerged adult individuals fed with different Passiflora biflora host plant treatment types (see Fig. 1A) as larvae. (Boxplot description as in A) (C) Parent-offspring regressions of total biosynthesized CNglcs concentration.

First, we consider the estimated broad-sense genetic variance component and estimate evolutionary potential of toxicity, assuming the genetic component consists of largely additive genetic variance. The broad-sense evolvability of total cyanogen concentration was eμ = 1.55%, SE ± 0.89 (eμ(linamarin) =1.89%, eμ(lotaustralin) =1.04%; Table 2, File S1), meaning that the trait mean is expected to change by 1.55% per generation per unit strength of selection (Hansen & Houle, 2008; Hansen, Pélabon & Houle, 2011).

Our baseline estimates of repeatability, as well as broad-sense heritability and evolvability (Model 1) thus showed that there is moderate but detectable variation among broods in biosynthesized toxicity. We then estimated maternal and paternal contributions to this variation. When including maternal identity as a second random effect in the animal model (Model2), this maternal variance component explained most of the variance explained by the genetic variance component in Model1, whereas VG became negligible, illustrating that the two random terms explain essentially the same variance component (i.e., the among-brood variance; Table 2, File S1). Parent-offspring regressions revealed that offspring cyanogen concentration was positively associated with the cyanogen concentration of the mother (slope =0.181 ± 0.071, t =2.539, P = 0.012 for total cyanogen concentration; Fig. 2C), while the association between fathers and offspring showed a negative, but statistically non-significant trend (slope = −0.122 ± 0.083, t =  − 1.469, P = 0.143 for total cyanogen concentration). In turn, the midparent–offspring regression was nearly flat (slope =0.046 ± 0.107, t =0.435, P = 0.664 for total cyanogen concentration; Fig. 2C). These regression slopes translate into a maternal-effect coefficient for total cyanogen concentration of m =0.261, CI = [−0.099, 0.614] (m (linamarin) =0.288, CI = [−0.094, 0.687]; m (lotaustralin) =0.193, CI = [−0.135, 0.520]), which implies the existence of positive maternal effects in biosynthesized cyanogenic toxicity. However, uncertainty in these maternal effects remains notable, as illustrated by the wide confidence intervals.

Finally, we investigated other sources, particularly environmental origins of variance. Larvae reared on different host plant types (Fig. 1A) did not differ in biosynthesized cyanogenic toxicity as adults (Fig. 2B), and the host plant type explained a negligible proportion of variance in butterfly toxicity level (Vtrm/VP = 0.011; Fig. 1D, Table 2, File S1). This is despite the strong effect of the watering treatments on the cyanogenic toxicity and quality indicators of the host plants (e.g., nitrogen-carbon balance; see File S3), and also indicates that among-brood variance is unlikely to be explained by common environment effects related to dietary quality. Butterfly sex was also a negligible source of variation in toxicity (Vsex/VP = 0.011; Fig. 1D, Table 2, File S1). Despite the common-garden conditions (outside of dietary treatments) that all individuals experienced throughout their life cycle, the residual (unexplained) variance component of cyanogen concentration remained large (Vres/VP = 0.874 for total cyanogens, Fig. 1D, Table 2, File S1). This large amount of unexplained variance is also demonstrated by a difference in toxicity level between the parental and offspring generation (F1,320 = 27.42, P < 0.001, effect size (Hedges’ g) =0.962 for total cyanogens), despite the two generations sharing common garden conditions.

Discussion

We have documented extensive variation in chemical defenses, both within and across, natural populations of H. erato. Wide variation in defensive toxins has often been found in chemically defended prey, yet the sources of this variation remains largely unclear (Speed et al., 2012). Our data shows that intraspecific variation in de novo synthesized chemicals was not strongly associated with environmental conditions and levels of biosynthesized toxicity were similar in butterflies collected across steep environmental gradients. Similarly, our experimental work consistently showed that variation in host plant biochemical profiles and nutritional quality do not contribute to variation in H. erato biosynthesized toxicity. Instead, the variation has a moderate but detectable broad-sense genetic component (12.5% of variation explained by brood), which could, given that it was largely composed of additive genetic effects, imply substantial evolutionary potential in biosynthesized chemical defenses of H. erato. Our broad-sense heritability estimate of biosynthesized cyanogenic toxicity (0.115 ± 0.086) is close to the average heritability value of physiological traits (0.12 ±0.05), based on a meta-analysis of a wide range of traits and taxa (Hansen, Pélabon & Houle, 2011). This estimate differs markedly, though, from that obtained in a recent study on toxicity in warningly colored wood tiger moths (Arctia plantaginis), in which no evidence for a genetic component to variation in the secretion amounts of de novo synthesized chemical defenses could be detected (Burdfield-Steel et al., 2018; Lindstedt et al., 2020). A few other studies have estimated genetic components of insect chemical defense variation, with some detecting a genetic component (Eggenberger & Rowell-Rahier, 1992; Holloway, De Jong & Ottenheim, 1993; Yezerski, Gilmor & Stevens, 2004) and others not (Müller et al., 2003). Our study adds on these previous work by investigating chemical defense variation across environments and mimicry rings.

Our estimated value for mean-scaled evolvability of cyanogenic toxicity, which gives a better standardized measure of evolutionary potential (Hansen & Houle, 2008; Hansen, Pélabon & Houle, 2011) is high (1.55 ± 0.89%), especially when compared to the average value of physiological traits (0.49 ± 0.14%), as well as the median evolvability of all studied traits across taxa (0.26 ± 0.03%; see Hansen, Pélabon & Houle, 2011). The evolvability of eμ = 1.55% means that the mean cyanogenic toxicity can be expected to change by 1.55% per unit strength of selection per generation. This high estimate of evolvability would imply that the trait has the potential to evolve rapidly, especially in conditions of strong selection. The effect would be magnified given the short generation time (usually one to two months) of these tropical butterflies. For example, assuming Heliconius has six generations per year, and increased cyanogenic toxicity is selected for with selection as strong as selection on fitness as a trait (βμ = 100%, see Hereford, Hansen & Houle, 2004), the trait mean would have the potential to increase by 9.7% in just one year.

However, our data from experimental crosses indicate that the inheritance of biosynthesized cyanogenic toxicity has a more complex basis, beyond pure additive genetic effects. Particularly, there is evidence for a central role for maternal effects. This is supported by our observations that (1) the maternal effect component fitted in our animal model seemed to explain the same component of variance as the broad-sense genetic component, as well as that (2) midparent–offspring and father-offspring regressions were non-significant while mother and offspring toxicity values were positively correlated. Given that mothers were raised in common garden conditions, and environmental variance is thus expected to be low, these effects likely have a genetic or epigenetic basis. At this point, it is unclear if the maternal effects that we document here are the result of maternally transmitted genes, epigenetic effects, direct transfer of toxins from mother to offspring in eggs (Nahrstedt & Davis, 1983; Winters et al., 2014), or other types of maternal influence on the offspring phenotype (Lande & Price, 1989; Lynch & Walsh, 1998b; Walsh & Lynch, 2018b). Females are the heterogametic sex in Lepidoptera, but gene content of the highly heterochromatic W chromosome is poorly explored. Heliconius eggs have very high concentrations of cyanogenic toxins (Nahrstedt & Davis, 1983; de castro et al., 2020) and the relationship between the provisioning of compounds in eggs by mothers and the level of these compounds in adults could be explored in future studies. We did not detect any paternal effects, even though Heliconius males are known to transfer a nuptial gift containing biosynthesized cyanogenic compounds to females at mating (Cardoso & Gilbert, 2007).

Independent of what is driving the maternal effects, the occurrence of a maternal effect of the type estimated here, defined simply as an effect of the mothers phenotype on the offspring’s phenotype (as in Falconer’s dilution model; Lande & Price, 1989; Lynch & Walsh, 1998a), could considerably strengthen the response of the trait to selection. Although there is much uncertainty in how maternal effects will affect the response to selection (Walsh & Lynch, 2018a), the predicted cumulative selection response with maternal effects under Falconer’s model for a trait with a maternal effect coefficient of m =0.25, which is comparable to the m =0.261 estimated here for cyanogenic toxicity, would over time exceed that of a trait with m = 0 by as much as 50% (assuming constant heritability) (Walsh & Lynch, 2018a). This effect, however, is expected to dilute over generations (Walsh & Lynch, 2018a). The maternal coefficient should also be interpreted with caution because of remaining uncertainty in our estimate. Predicting the response of toxicity to selection in the wild could also be complicated by the observed skewed distribution of toxicity, as skewed breeding values could generate an asymmetrical selection response (Bonamour et al., 2017). Furthermore, we cannot rule out that some of the observed among-brood variance is explained by common environment effects other than maternal effects and the measured effect of diet, e.g., differences in group size during larval development. However, the demonstrated negligible plastic responses to diet, common-garden rearing, and the temporal spread of egg-laying of each brood (eggs laid singly over a time period of up to two weeks per brood) are expected to reduce the likelihood of such common environment effects among broods.

Implications of evolutionary potential in chemical defenses for aposematic and mimetic systems

Studies of mimicry often focus on color pattern, i.e., the warning signal (Benson, 1972; Mallet & Barton, 1989; Kapan, 2001; Merrill et al., 2015; Jiggins, 2017). However, it is important to note that contrasting selective agents could be acting on the defense trait associated with the warning signal, both traits equally required for aposematism and mimicry. The selection pressures acting on chemical defense traits can be expected to be complex and dynamic (Speed et al., 2012; Jones, Speed & Mappes, 2016; Briolat et al., 2019). The role of predators as the major selective agent is clearly a key component of possible adaptive evolutionary processes herein (Ruxton et al., 2018; Briolat et al., 2019). Even high evolutionary potential will only lead to evolutionary changes if the defense trait variation influences predation rates (Speed et al., 2012). In general, empirical data on the impact of variation in defenses and its signaling on predators is lacking, but there is evidence for variation among predators in traits including perceptual abilities, individual motivation and tolerance to chemical defenses (Halpin et al., 2017; Briolat et al., 2019; Hämäläinen et al., 2020). Predators may also respond not only to average defense levels, but to the variation in them (Skelhorn & Rowe, 2005; Barnett, Bateson & Rowe, 2014). Furthermore, prey toxicity level does not necessarily correlate with actual unpalatability, the ultimate cue for predator choices (Winters et al., 2018). Predator-mediated selection is therefore expected to be dynamic and difficult to estimate. In addition, bottom-up selection related to co-evolutionary relationships with host plants, as well as selection on energy optimization may also play a role (Turner, 1984; Ruxton et al., 2018).

Species may only respond adaptively to such selection pressures given heritable genetic variation in these traits. Evolutionary potential in chemical defenses could allow species to respond to the expected dynamic top-down and bottom-up selection, resulting in diverse patterns of adaptive evolution in chemical defense traits. This could have important consequences for the evolutionary trajectories of aposematic and mimetic species, and ultimately play a role in the evolution of warning signal diversity. This is because evolutionary changes in chemical defense levels could alter mimetic relationships among groups of species across geographic space, including shifts between Müllerian, Batesian and quasi-Batesian relationships, which could in turn influence advergent vs. convergent selection and contribute to the observed diversity in mimicry systems (Mallet & Joron, 1999; Speed, 1999; Briolat et al., 2019). With regards to Heliconius, the combination of a locally dynamic selective environment, flexible genetic mechanisms for generating warning signals (Wallbank et al., 2016; Van Belleghem et al., 2017), and evolutionary potential in toxicity could explain the rapid origin of inter- and intraspecific variation. In this light, the ability of Heliconius to biosynthesize defense compounds could allow for increased genetic control of chemical defenses compared to most other chemically defended Lepidoptera, which rely solely on host plant compound sequestration (Bowers, 1992; Nishida, 2002; Zvereva & Kozlov, 2016). Because Heliconius can sequester cyanogenic compounds of only some Passiflora species (Engler, Spencer & Gilbert, 2000; de castro et al., 2019), selection would be targeted toward de novo synthesized defenses particularly in environments lacking such host species. Furthermore, previous investigations both confirmed that the cyanogenic glucoside compounds in Heliconius deter vertebrate predators (Cardoso, 2020) and that predator avoidance learning rate increases when comparing low to moderate levels of cyanogenic compound concentrations (Chouteau et al., 2019). Thus, the level of cyanogen defenses is likely to be a direct target of selection by predators in some circumstances.

By altering the nature of mimetic relationships or selection for increased defense levels on a new warning signal form, evolvability of defense traits could drive the evolution and spread of new color patterns and the reproductive isolation that may follow, potentially even leading to the formation of new species (Mallet, McMillan & Jiggins, 1998; Jiggins et al., 2001). A phylogenetic signal in the cyanogenic glucoside chemical profile in the Heliconiini tribe supports this idea (de castro et al., 2019; Sculfort et al., 2020) and underscores the potential for adaptive evolution in chemical defense traits to influence evolutionary trajectories in mimicry systems. However, more empirical data is called for in future studies to estimate levels of evolutionary potential and patterns of selective responses in defensive traits in aposematic and mimetic species.

Evolutionary processes shaping chemical defense variation: the role of automimicry

A further complication in chemical defense evolution is the interplay between individual and public good (Jones, Speed & Mappes, 2016). In this respect, our data demonstrating a broad-sense genetic component in the defense trait coupled with patterns of natural variation of cyanogen levels in Heliconius erato populations are consistent with balancing selection related to automimicry, in which automimic “cheaters” exploit the protection given by their warning coloration with investing less in chemical defenses themselves. Automimicry may occur if producing and maintaining chemical defenses is costly (Brower, Pough & Meck, 1970; Speed et al., 2012), leading to selection reducing individual-level toxicity (Bowers, 1992; Zvereva & Kozlov, 2016; Ruxton et al., 2018), and if there are also individual benefits of investing in chemical defenses (e.g., Ruxton & Speed, 2006; Svennungsen & Holen, 2007). Previous studies indicate that de novo synthesis of defenses may generally be more costly than the sequestration of plant compounds (Zvereva & Kozlov, 2016), which may make automimicry more likely in species with biosynthesized defenses. The occurrence of automimicry could lead to an increase in population average palatability, which could theoretically dilute the protection conveyed by the warning signal (Gibson, 1984; Speed et al., 2012; Jones, Speed & Mappes, 2016), as in Batesian and quasi-Batesian mimicry, and influence the evolutionary trajectories of the species sharing the warning signal. Thus, empirical data on within-population defense variation and automimicry can give important insights into the diversity of mimetic systems (Jones, Speed & Mappes, 2016).

Our results show that all three butterfly populations along the studied rainfall gradient exhibited similar skewed distributions of biosynthesized cyanogens. The majority of individuals in these populations had relatively low, although not zero, levels of toxicity (24% of individuals had less than 0.5% concentration of biosynthesized cyanogenic compounds from an overall range of 0.1–2.9%), whereas a smaller proportion had substantially higher levels, and the pattern was surprisingly similar in all populations. Such a pattern could be explained by age distributions (Heliconius have maximum adult lifespans in excess of six months and continuous reproduction throughout life; Gilbert, 1972; Dunlap-Pianka, Boggs & Gilbert, 1977), as previous studies suggest that Heliconius accumulate cyanogenic toxins with age (Nahrstedt & Davis, 1985; de castro et al., 2020), or by individual differences in the balance between sequestration and biosynthesis of cyanogenic compounds (Engler-Chaouat & Gilbert, 2007; de castro et al., 2019; de castro et al., 2020; de castro et al., 2021). More generally, chemical defense polymorphisms could also be maintained by patterns of predator selection targeting defense variation itself (Skelhorn & Rowe, 2005; Barnett, Bateson & Rowe, 2014). However, as many individuals in the study populations had near zero levels of biosynthesized toxins, the pattern could also be indicative of automimicry. In Heliconius, the costs of chemical defense remain poorly known, but some cases of automimicry have previously been reported (Arias et al., 2016). Here, toxicity and mass were not associated, which could imply that potential costs of cyanogen biosynthesis are not related to or mediated by a trade-off with body size. In terms of benefits of chemical protection, requirements for automimicry are likely to be met, as empirical data on Heliconius imply direct benefits (surviving predator attacks) for chemically defended individuals (Boyden, 1976; Pinheiro, 1996; Pinheiro & Campos, 2019). Furthermore, an increase in protection level is found especially when comparing low to moderate cyanogenic toxicity levels (Chouteau et al., 2019), applicable to the observed cyanogen levels of H. erato (Arias et al., 2016; de castro et al., 2019). However, future studies on natural defense variation and automimicry in Heliconius should measure the entire range of cyanogens (both sequestered and biosynthesized) and other chemical defense compounds potentially interacting to produce the unpalatability level perceived by relevant predators (Jiggins, 2017; Winters et al., 2018). In more general terms, future research in this framework should investigate the costs of defensive compound biosynthesis (see also Zvereva & Kozlov, 2016) and its role in the evolution of automimicry, as well as automimicry-originating defense variation as a potential source introducing diversity into mimetic systems.

The roles of phenotypic plasticity and ecological sources of defense variation in wild populations

It has been suggested that the often wide variation in defense toxins could be explained by the non-adaptiveness of toxicity profiles, in which case the variation would stem from chance effects and plastic responses to the environment (Speed et al., 2012). Empirical data on chemical defense variation in wild populations e.g., along environmental gradients could help to evaluate ecological vs. evolutionary origins of chemical defense variation, and the expected consequences for diversity in aposematic and mimetic systems (Briolat et al., 2019). Although the large amount of unexplained variance in our quantitative genetics model may suggest the presence of unknown environmental sources of variation in biosynthesized toxicity, we found very little evidence for a role of known ecological factors in explaining variation in toxicity profiles along our studied environmental gradients, despite the steepness of these gradients. For example, the chemical profiles were very similar in the Panamanian Wet and Dry populations, which experience a five-fold difference in rainfall. Similarity among populations is unlikely to be caused by large-scale dispersal along the gradient, because Heliconius erato has a restricted home range and a dispersal distance of around 1–2 km per generation (Turner, 1971). For the same reason, the collection locations of adults are also likely to reflect the locations in which larvae fed. Furthermore, despite water availability inducing substantial differences in the cyanogen content and quality indicators of P. biflora, both our field and experimental data indicate that variation within the host plant species does not contribute to variation in H. erato biosynthesized toxicity. This is in contrast to our expectations of dietary quality as a major source of plastic chemical defense variation.

However, there were striking differences in the chemical profiles between the two distinct warning color forms of H. erato that we sampled from geographically distant populations in Panama and Ecuador. Compared to their Panamanian conspecifics sampled along the gradient of rainfall at sea level, the Ecuadorian populations inhabit a very different and geographically distant biotope along the eastern slope of the Andes (Montejo-Kovacevich et al., 2020). These environments differ in larval host plant and nectar/pollen availability, the community of predators and of co-mimics, which are far more diverse in Ecuador. Lower levels of cyanogen biosynthesis in Ecuadorian populations of H. erato may suggest that these populations have specialized on a Passiflora host which provides more cyanogens for sequestration (Engler-Chaouat & Gilbert, 2007; de castro et al., 2019; de castro et al., 2020; de castro et al., 2021). Alternatively, the more diverse mimetic ring in Ecuador could also provide greater protection by more toxic model species, relaxing selection on defenses in co-mimics (Turner, 1987; Speed, 1993; Speed, 1999; Mallet, 1999). Differences in the collection methods between populations in Panama and Ecuador are unlikely to explain the extremely low levels of biosynthesis in the Ecuadorian individuals, as biosynthesis levels in common-garden reared individuals are generally shown to be similar with the levels of corresponding wild individuals in the current and previous studies on Heliconius (Engler-Chaouat & Gilbert, 2007; de castro et al., 2020).

In this light, our data may provide an example of how ecological variation could lead to differential selective environments between conspecific populations, thereby leading to local adaptation of the acquisition mechanisms and levels of defensive compounds. Such adaptive radiations would be facilitated by evolutionary potential in defense traits. More data on different aposematic and mimetic systems will be needed to confirm whether ecological variation may have a greater influence on chemical defense variation through introducing spatially and temporally varying selection pressures, rather than by introducing “ecological noise” through plastic responses in defense traits.

Conclusions

The role of ecological and evolutionary processes in the origins and maintenance of the wide variation in chemical defenses, and the consequences for the evolution of aposematism and mimicry holds many unanswered questions. Our analysis of biosynthesized cyanogenic toxicity in Heliconius erato, studied along natural environmental gradients and in a common garden, shows that this important chemical defense trait varies substantially within and among populations. We found no evidence for a role of dietary variation (within host species) or steep environmental gradients in explaining toxicity profiles, despite a large component of unexplained variance in common-garden reared broods. Instead, the results suggest that adaptive processes originating from e.g., larval host species availability, mimicry environment, and individual-level energy optimization may play important roles in the variation of cyanogenic defenses. This is in line with our data indicating substantial broad-sense evolvability in cyanogenic toxicity, although more data will be needed to confirm the origins of the observed among-brood variance and shed light on how the indicated maternal effects will affect the overall selection response. Our study suggests that adaptive evolution could be an important force driving variation in defensive traits. Such adaptive processes could play a part in explaining the incredible diversity in aposematic and mimetic systems.

Supplemental Information

Supplemental Information 1 A. Full results table of between-population variation of cyanogenic glucoside compounds in natural populations of Heliconius erato and Passiflora biflora. B. Full results table of REML animal models

Click here for additional data file.

Supplemental Information 2 Summary table of Heliconius erato broods

Click here for additional data file.

Supplemental Information 3 Cyanogenic toxicity and overall quality of host plant Passiflora biflora watering treatment groups

Click here for additional data file.

Supplemental Information 4 Detailed information about methods used in the study

Click here for additional data file.

Supplemental Information 5 Cyanogenic glucoside concentration and associated data for common garden Heliconius erato individuals

Click here for additional data file.

Supplemental Information 6 Cyanogenic glucoside concentration and associated data for field-collected Heliconius erato individuals

Click here for additional data file.

Supplemental Information 7 Pedigree data associated with common garden Heliconius erato individuals in Data S1

Click here for additional data file.

Supplemental Information 8 Cyanogenic glucoside concentration and associated data for common garden Passiflora biflora samples

Click here for additional data file.

Supplemental Information 9 Cyanogenic glucoside concentration and associated data for field-collected Passiflora biflora samples

Click here for additional data file.

Personnel and researchers at the Smithsonian Tropical Research Institute (STRI Panama) and Ikiam at Universidad Regional Amazónica (Ecuador) are acknowledged for help with permits and the field collection of the study species. We are especially grateful to Marleny Rivera (STRI) for reliable and friendly assistance in field sampling. Samples were processed at the Molecular Ecology Laboratory at the University of Helsinki (UH) and analysed at the HiLIFE NMR facility at the Institute of Biotechnology (UH), a member of Instruct-ERIC Centre Finland, FINStruct, and Biocenter Finland. Søren Bak and David Pattison (Department for Plant and Environmental Sciences, University of Copenhagen) are acknowledged for collaboration on the LC-MS analyses of Passiflora samples. Juha-Matti Pitkänen, Tuomas Niemi-Aro and Annukka Ruokolainen are thanked for technical assistance, and Guillaume Minard, Anne Duplouy, Richard Wallbank, Richard Merrill, Oscar Paneso, Krzysztof Kozak and Sophia Gripenberg for advice on study design and lab and field work. Mika Zagrobelny and Marcio Cardoso are thanked for advice on cyanogenic assays and Mika Zagrobelny also for providing reference samples of the CNglc compounds.

Additional Information and Declarations

Competing Interests

Author Contributions

Field Study Permissions

Data Availability

The authors declare there are no competing interests.

Anniina L.K. Mattila conceived and designed the experiments, performed the experiments, analyzed the data, prepared figures and/or tables, authored or reviewed drafts of the paper, and approved the final draft.

Chris D. Jiggins conceived and designed the experiments, authored or reviewed drafts of the paper, and approved the final draft.

Øystein H. Opedal analyzed the data, authored or reviewed drafts of the paper, and approved the final draft.

Gabriela Montejo-Kovacevich and Érika C. Pinheiro de Castro performed the experiments, authored or reviewed drafts of the paper, and approved the final draft.

W. Owen McMillan and Caroline Bacquet performed the experiments, authored or reviewed drafts of the paper, provided field-work facilities and project administration, and approved the final draft.

Marjo Saastamoinen conceived and designed the experiments, performed the experiments, authored or reviewed drafts of the paper, and approved the final draft.

The following information was supplied relating to field study approvals (i.e., approving body and any reference numbers):

Research and collecting permits were granted by the Ministerio de Ambiente, Republica de Panama (Permit SE/AP-21-16) and the Ministerio del Ambiente, Ecuador (under the Contrato Marco MAE-DNB-CM-2017-0058).

The following information was supplied regarding data availability:

All data generated and analyzed during this study are available in the Supplementary Files.

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
