# Peer review of "Evolutionary and ecological processes influencing chemical defense variation in an aposematic and mimetic Heliconius butterfly"

_PeerJ, doi:10.7717/peerj.11523_

## Round 0.1 · original submission · Minor Revisions

We have received two reviews for your manuscript and both reviewers were positive about your study. Both reviewers suggested that some additional background/context details about the populations are needed. Reviewer 1 raised minor points related to clarifications of the experimental design and Reviewer 2 suggested additional references that should be integrated and discussed in the manuscript. The resolution of figure 1 needs to be improved and figure 2b was missing.

In addition to these comments, I would also suggest adding a power analysis for your animal model analysis, to further support the validity of the results presented here.

I suggest the approach used by Class, B., & Brommer, J. E. (2020). Can dominance genetic variance be ignored in evolutionary quantitative genetic analyses of wild populations?. Evolution, 74(7), 1540-1550.

See also Bourret, A., & Garant, D. (2017). An assessment of the reliability of quantitative genetics estimates in study systems with high rate of extra-pair reproduction and low recruitment. Heredity, 118(3), 229-238.

·

Basic reporting

This is a very well written manuscript. The introduction provides an excellent explanation and introduction to chemical defense variation and mimicry, especially in butterflies. The authors clearly lay out their research questions at the end of the introduction section. The discussion section, however, is too long and should be shortened considerably.

The figures are relevant, but in this version of the manuscript they were not of sufficient quality. The lines were fuzzy and the text had shadows. I was confused by Figure 2B, as it appears to be blank on my copy of the manuscript. The raw data are shared.

Experimental design

The experimental design is well described in the main text and the supplementary material. I, however, have a few questions and some suggestions to improve the clarity of the findings.

1. How is it possible to chemically distinguish the cyanogens that were sequestered vs. synthesized?
2. What is the reasoning behind preservation in MeOH instead of EtOH?
3. The collection procedure for the plant leaves needs to be justified. Usually, plant material is preserved instantly on dry ice or in liquid nitrogen to preserve phytochemical properties. What compounds are being targeted using this type of preservation method?

L102: Should read: …were separated by sex, weighed, and….
L105: How were individuals tracked?
L106: Sampled is not the right word here…maybe stored or preserved?
L110: Specify that cuttings of Passiflora biflora were taken for cultivation?
L126: Should read: …slightly drooped between watering events.
L155: Insert version and date of software used.
L183: It is not clear what the average n refers to here.
L189: Need subject and verb agreement here: concentration…was or concentrations….were.

Validity of the findings

The findings are well presented and the discussion is robust without over speculation. The discussion is too long and should be made more succinct. I have a few questions about results and perhaps, if the suggestions are valid, they could be included as another potential explanation. I also have a few further requests for clarification in the writing.

1. Is it possible that adult dispersal is occurring along the gradient and collection location of the adult does not reflect the location in which larvae fed?

2. How long do these butterflies live? What could be the extent of age distributions? Could variation also be due to the presence of post-reproductive individuals in the sample?

L297: I am not really clear about the animal model…have I missed something?
L336: Specify butterfly sex rather than plant sex.
L403: Provide an explanation and/or a reference for the dilution effect of the maternal effect.
L475: It would make sense to use “increase in palatability” instead of the double negative “decrease in unpalatability”.

Additional comments

This is an extremely interesting study that shows the extreme complexity of chemical defense evolution in this group of insects. It was a pleasure to read. I support acceptance of the manuscript...but I also would suggest very slight modifications as described above.

Reviewer 2 ·

Basic reporting

Literature references, sufficient field background/context provided: The authors do a good job highlighting the need for the study, and I wholeheartedly agree! However, some additional background/context and literature references could help fit the study within the broader field of knowledge and make the findings relevant and interesting for a broader readership. Specifically, regarding to the author's main question "what drives the evolution in defensive toxins within and among prey?"

It would be helpful for readers who are less familiar to the study system for the authors to expand this section beginning Page 8 line 78 "Heliconius erato provides an interesting study subject in terms of variation in chemical defenses, warning coloration patterns and mimicry relationships, as it has a diverse array of color forms associated with local color pattern mimicry rings, and is found in ecologically diverse habitats across a large geographic range." Missing citations. Can the authors expand how the colour forms vary and how they are associated with local mimicry rings? This background information is necessary to fully understand differences in selection between the populations of the study but it is lacking.
Highly relevant to this manuscript but not mentioned, is how predators respond to variation in chemical defences. I would love to see the authors discuss how variation itself can be beneficial:
Skelhorn, John, and Candy Rowe. "Frequency-dependent taste-rejection by avian predation may select for defence chemical polymorphisms in aposematic prey." Biology Letters 1.4 (2005): 500-503.
Barnett, Craig A., Melissa Bateson, and Candy Rowe. "Better the devil you know: avian predators find variation in prey toxicity aversive." Biology letters 10.11 (2014): 20140533.
Also relevant but not mentioned, is the potential for variable defences to be maintained through warning signal honesty.
Blount, Jonathan D., et al. "Warning displays may function as honest signals of toxicity." Proceedings of the Royal Society B: Biological Sciences 276.1658 (2009): 871-877.
Blount, Jonathan D., et al. "How the ladybird got its spots: effects of resource limitation on the honesty of aposematic signals." Functional Ecology 26.2 (2012): 334-342.
Arenas, Lina María, Dominic Walter, and Martin Stevens. "Signal honesty and predation risk among a closely related group of aposematic species." Scientific reports 5.1 (2015): 1-12.
Vidal-Cordero, J. M. et al. Brighter-colored paper wasps (Polistes dominula) have larger poison glands. Front. Zool. 9, 20 (2012).
Bezzerides, A. L., McGraw, K. J., Parker, R. S. & Husseini, J. Elytra color as a signal of chemical defense in the Asian ladybird beetle Harmonia axyridis. Behav. Ecol. Sociobiol. 61, 1401–1408 (2007).
Cortesi, F. & Cheney, K. L. Conspicuousness is correlated with toxicity in marine opisthobranchs. J. Evol. Biol. 23, 1509–1518 (2010).
Page 23 Line 425: "...but there is evidence for variation among predators in traits including perceptual abilities, individual motivation and tolerance to chemical defences." Suggested additional citation: Hämäläinen, Liisa, et al. "Predators’ consumption of unpalatable prey does not vary as a function of bitter taste perception." Behavioral Ecology 31.2 (2020): 383-392.
On the role of unequal defences in mimicry rings, this notable exception where unequal defences can be mutualistic was excluded:
Rowland, Hannah M., et al. "Co-mimics have a mutualistic relationship despite unequal defences." Nature 448.7149 (2007): 64-67.
On the role of maternal effects:
Winters, Anne E., et al. "Maternal effects and warning signal honesty in eggs and offspring of an aposematic ladybird beetle." Functional Ecology 28.5 (2014): 1187-1196.

Professional article structure, figures, tables. Raw data shared.
Figure 1 is not of sufficient resolution, small text is illegible.
Figure 1A: In the description, "... in total- and total biosynthesized cyanogenic (CNglc) toxin concentrations..." Is this a typo? What is "total-" ? According to the x-axis label, it is % that is reported rather than the total that is reported. The plant seems to be reported in ug/mg while the butterfly is in %. Why the difference? Can both be in %? Information about the type of plot and error bars is missing.
Figure 1D: Information about the type of plot and error bars is missing.
Figure 2A information about the type of plot and error bars is missing.
Figure 2B is an empty/blank plot.

Experimental design

no comment

Validity of the findings

On the question: Can some of this variation be explained by environmental gradients or highly divergent habitat types across a wider geographic scale?
In the results, the Ecuadorian population has significantly lower levels of synthesised cyanogens than the Panamanian populations. In the discussion, the authors speculate that this difference might be due to specialisation on a different host plant that provides more cyanogens for sequestration, or alternatively that the more diverse mimicry ring could provide greater protection by a more toxic model species. I wonder if differences in collection and sampling between the two locations could explain the difference? In Panama, wild-caught individuals were used and the sample for analysis included the "body excluding wings and one half of the thorax." However, in Ecuador, the F1 generation was used after being raised in a common garden environment and the sample for analysis included only the thorax. Therefore alternative explanations for the lower levels of cyanogens in the Ecuadorian population could be that 1) Synthesis was not as efficient in the common garden experiment compared to wild-caught individuals 2) cyanogens are stored in higher concentrations in the body parts that were sampled in Panamanian individuals but not form Ecuadorian individuals i.e. head, abdomen and legs. Can the authors discuss this possibility?

Additional comments

This paper was a pleasure to read. It is well-written and easy to follow and fills an important knowledge gap. Some additional background/context and literature references could help fit the study within the broader field of knowledge and make the findings relevant and interesting for a broader readership. I have concerns about comparing Ecuadorian and Panamanian samples with different collection methods, which I described in more detail in the relevant section.

---

## Round 0.2 · accepted · Accept

I am pleased with the revisions made on the manuscript.